# Implications of Kynurenine Pathway Metabolism for the Immune System, Hypothalamic–Pituitary–Adrenal Axis, and Neurotransmission in Alcohol Use Disorder

**DOI:** 10.3390/ijms25094845

**Published:** 2024-04-29

**Authors:** Bartosz Osuch, Tomasz Misztal, Kinga Pałatyńska, Dorota Tomaszewska-Zaremba

**Affiliations:** Department of Animal Physiology, The Kielanowski Institute of Animal Physiology and Nutrition, Polish Academy of Sciences, Instytucka 3, 05-110 Jabłonna, Poland; t.misztal@ifzz.pl (T.M.); k.palatynska@ifzz.pl (K.P.); d.tomaszewska@ifzz.pl (D.T.-Z.)

**Keywords:** alcohol use disorder, kynurenine pathway, immune system, hypothalamic–pituitary–adrenal axis

## Abstract

In recent years, there has been a marked increase in interest in the role of the kynurenine pathway (KP) in mechanisms associated with addictive behavior. Numerous reports implicate KP metabolism in influencing the immune system, hypothalamic–pituitary–adrenal (HPA) axis, and neurotransmission, which underlie the behavioral patterns characteristic of addiction. An in-depth analysis of the results of these new studies highlights interesting patterns of relationships, and approaching alcohol use disorder (AUD) from a broader neuroendocrine–immune system perspective may be crucial to better understanding this complex phenomenon. In this review, we provide an up-to-date summary of information indicating the relationship between AUD and the KP, both in terms of changes in the activity of this pathway and modulation of this pathway as a possible pharmacological approach for the treatment of AUD.

## 1. Introduction

Tryptophan (TRP), an exogenous amino acid essential for the proper functioning of the body, is not only a compound included in proteins but also a precursor for the synthesis of various biologically active molecules. Enzymatic pathways metabolize the vast majority of TRP. The main non-protein pathway of TRP catabolism is the kynurenine pathway (KP), which forms several metabolites involved in many different, often opposing, physiological processes [1]. In general, the KP is the starting point for the synthesis of nicotinamide adenine dinucleotide (NAD), involved in the course of oxidation-reduction (redox) reactions crucial for cell signaling (including underlying biological processes related to signal transduction mechanisms in neurons) and cellular homeostasis involved in the maintenance of energy metabolism (NAD plays an essential role in the course of metabolic reactions that result in the production of adenosine 5′-triphosphate (ATP)), cell cycle regulation, immune response, and modulation of cell growth and apoptosis [2,3]. However, more detailed observations of individual KP metabolites have revealed that their systemic functions may be much broader. As a result, numerous in-depth studies have been conducted on the role of KP metabolites in physiological and pathological processes in the central nervous system (CNS) [4]. Consequently, it has become possible to examine their importance in the onset and course of psychiatric and neurological diseases [5].

The most common mental disorder is alcohol use disorder (AUD), a condition characterized by an impaired ability to stop or control alcohol use despite adverse social and health consequences. Persistent changes in the KP caused by alcohol abuse are suspected to perpetuate AUD and make individuals vulnerable to relapse [6,7]. Recent discoveries have shown clear implications of the KP for AUD [8]. There is no doubt that the KP is an essential link between the nervous and immune systems. Metabolites formed along the TRP-KP have attracted considerable interest, as it is beginning to be understood that they are at the interface of the relationship among the endocrine system, immune response, and brain function [9,10]. These discoveries have set new targets for neurobiological and pharmacological research. These goals are aimed, among other objectives, at finding ways to exploit the therapeutic potential of the pathway’s metabolites, as this provides a starting point for the synthesis of essential neurotransmitters. Moreover, several transient products of the path are modulators of important receptors while also influencing inflammatory responses and changes in the oxidative-reduction potential of neural cells [11,12]. This indicates the need to study the relationship between the KP and complex neurobiological and immunological mechanisms. Understanding the pathomechanisms of AUD and a thorough evaluation of the effects of ethanol (EtOH) on brain function will allow the development of new effective therapeutic interventions. Thus, clinicians and basic science researchers should collaborate to seek interrelationships between neurobiological research and specific clinical aspects of AUD.

## 2. Tryptophan Metabolism

TRP is an amino acid that undergoes complex metabolic pathways to form numerous intermediate compounds. TRP can be metabolized through two main pathways: the one that proceeds with the retention of an indole ring, and the other initiated by oxidative cleavage of the indole ring. Maintaining intact the indole ring of TRP leads to synthesizing bioactive indole compounds, including serotonin as a neurotransmitter and melatonin as a neuromodulator. On the other hand, oxidative cleavage of the indole ring initiates the KP, limiting TRP’s availability to synthesize these neurotransmitters [13,14] (Figure 1).

### Kynurenine Pathway (General Characteristics)

The first step in the KP, which involves breaking the indole ring of TRP, is catalyzed by enzymes indoleamine 2,3-dioxygenase (IDO) and tryptophan 2,3-dioxygenase (TDO). IDO and TDO exhibit differential expression patterns that influence the diversity of their functions depending on their localization in mammalian cells and tissues [15]. While IDO expression ensures local degradation of TRP in peripheral tissues and is limited to peripheral sites of immune modulation, TDO is massively expressed in the liver and is responsible for 90% of TRP catabolism. Specific pro-inflammatory cytokines, such as interferon-gamma (IFN-γ), tumor necrosis factor alpha (TNF-α), and the interleukins (IL)-1β and IL-6, can stimulate cell IDO synthesis [16]. On the other hand, TDO activity increases in response to an increase in blood glucocorticosteroid levels [17,18,19].

The intermediate product of the reactions catalyzed by TDO/IDO is N-formyl-L-kynurenine, which is immediately hydrolyzed by the enzyme kynurenine formamidase to the first stable intermediate metabolite—L-kynurenine (KYN) [20]. Kynurenine (KYN) is not neuroactive; however, it is broken down by three specific pathways to produce various neuroactive metabolites—some KYN metabolites are considered neuroprotective, while others are neurotoxic [1].

KYN is broken down to 3-hydroxy-kynurenine (3HK) by the enzyme kynurenine-3-monooxygenase (KMO). 3HK is then cleaved by kynureninase (KYNU), resulting in 3-hydroxyanthranilic acid (3-HANA). The next step involves the oxidation of 3-HANA to 2-amino-3-carboxymucono-6-semialdehyde (ACMS) by 3-hydroxy anthranilic acid oxidase (3-HAO). ACMS can be degraded through various transformations, leading to small amounts of neuroprotective picolinic acid and neurotoxic quinolinic acid (QUIN), ultimately degrading to the essential cofactor nicotinamide adenine dinucleotide (NAD^+^). KYN can also be degraded by kynureninase (KYNU) to form anthranilic acid, which is further degraded to 3HANA. In addition, it is also possible to convert KYN into neuroprotective kynurenic acid (KYNA) via irreversible transamination by kynurenine aminotransferase (KAT) [21,22].

## 3. Inflammation and Implications of Kynurenine Pathway Metabolism

In response to the toxic effects of EtOH and its metabolites, neuronal cell nuclei undergo gradual, systematic shrinkage and slow degradation. Damage to neural tissue is accompanied by an increase in the expression of genes responsible for microglial activation, and microglia cells proliferate and migrate to the site of damage. Activation of microglia is also associated with an increase in the level of factors with immune function, surface receptors that accelerate the immune response, and the synthesis of many compounds of a pro-inflammatory nature (including cytokines and chemokines). From a physiological point of view, this process is beneficial and aims to remove the harmful factor from the human system and re-establish a state of homeostasis. This phenomenon turns out to be pathophysiological when it occurs chronically. Patients who abuse alcohol have been shown to have chronically increased expression of pro-inflammatory cytokines in their serum and brain, as confirmed by numerous preclinical studies using animals. Particularly sensitive to inflammatory factors are the amygdala and hippocampus—brain structures responsible for emotion, memory, and cognitive function, as well as structures that are part of the reward system [23,24,25,26]. 

In response to chronic inflammation, cells synthesize IDO. The increase in IDO activity comes at the expense of a loss of TRP (by redirecting its metabolism toward the KP), to which T lymphocytes and natural killer (NK) cells are susceptible [27,28]. The consequence is cell cycle blockade in the G1 phase, inhibition of proliferation of these cells, and a state of immune tolerance [16,29,30]. In greater detail, this is because, in a TRP-deficient situation in the cell, transfer RNAs (tRNAs) cannot be aminoacylated, resulting in the filling of A sites of ribosomes with amino-deacylated tRNAs and stopping overall translation. Indeed, the absence of TRP in T cells leads to the activation of serine/threonine-protein kinase general control nonderepressible 2 (GCN2), which binds free, non-amino-acid-bound tRNA molecules, thus leading to the induction of integrated stress response (ISR). This results in the inhibition of protein biosynthesis on the matrix of messenger RNA (mRNA), which is synonymous with the inhibition of translational processes [31,32]. This effect is further enhanced because the first stable metabolite of the KP—KYN—can then be metabolized to further products of this pathway (3HK, 3HANA, QUIN) that exhibit cytotoxic effects, including conventional effector T cells [33]. The association of KP metabolites with neuroinflammation has been demonstrated. Interesting clinical results in this regard were presented by Yan et al. [34], showing a significant increase in neopterin, KYNA, QUIN, and the ratio of KYN to TRP (KYN/TRP) in the brain fluid of patients with inflammation. Neopterin, KYNA, KYN/TRP, and QUIN showed the best sensitivity in determining neuroinflammation. The results of this study suggest that KP metabolites (KYN, KYNA, and QUIN) are useful diagnostic biomarkers for monitoring neuroinflammation and provide insight into the role of the related inflammatory metabolism pathway.

In addition, inflammation leads to increased production of corticosteroids (glucocorticosteroids, i.e., cortisol, corticosterone, cortisone)—anti-inflammatory steroid hormones produced by the adrenal cortex under the influence of pituitary adrenocorticotropic hormone (ACTH). The mechanism of action of glucocorticosteroids is complex and not fully understood. However, we have a large body of evidence linking addiction to the endocrine system, which coordinates their secretion, creating structural and functional interconnections between the hypothalamus, pituitary gland, and adrenal glands (HPA axis) [35]. Chronic EtOH intake has been shown to disrupt the activity of the HPA axis and cause over-secretion of steroid hormones, resulting in the impairment of many physiological processes [36,37,38]. 

## 4. HPA Axis and Implications of Kynurenine Pathway Metabolism

Activation of the HPA axis originates in the small-cell neurosecretory neurons of the paraventricular nucleus of the hypothalamus, which are stimulated to secrete corticotropin-releasing hormone (CRH) when exposed to a stressor [39]. Interestingly, CRH is synthesized by neurons in which serotonergic fibers have terminations [40]. Thus, a dynamic interplay between serotonin neurotransmission and HPA axis activity was observed [41]. This is one of the reasons why dysfunction of the HPA axis (and related neurotransmitter systems) may be related to the occurrence of disorders and diseases that often coexist with addiction—including affective disorders or post-traumatic stress disorder [42]. Alcohol-dependent individuals have been shown to have HPA axis dysfunction that persists after withdrawal and during prolonged abstinence, which may be partly responsible for the onset of abstinence symptoms and relapse [43,44,45].

The neurohormone CRH then reaches the anterior lobe of the pituitary gland via the capillaries, where it stimulates the production and secretion of ACTH. In turn, ACTH enters the adrenal cortex cells with the blood, stimulating them to produce glucocorticosteroids. This entire system is subject to a mechanism referred to as a negative feedback system since an increase in glucocorticoids leads to inhibition of hypothalamic and pituitary activity, resulting in inhibition of further glucocorticoid release from the adrenal glands. In other words, CRH activates the secretion of ACTH, which then releases cortisol, and in turn, excess cortisol reciprocally inhibits CRH and ACTH secretion. In this way, the stress response is stopped [46,47].

Under physiological conditions, the HPA axis, by regulating the secretion of glucocorticosteroids, is tasked with preparing the body for a fight-or-flight response and is also involved in regulating emotions, mood, and sexual behavior. However, prolonged stress causes chronic elevation of glucocorticosteroids and disrupts the normal regulation of the HPA axis, which obviously translates into a pattern of neurocognitive impairment [48,49,50]. Stress-induced changes in the prefrontal cortex are thought to be responsible for the transition from independent drinking to addiction. EtOH addiction is associated with dysregulation of the HPA axis. A rat model of repeated EtOH intoxication–withdrawal cycles using chronic intermittent ethanol EtOH vapor inhalation (CIE) and abstinence from CIE showed a peak increase in plasma corticosterone levels during the CIE procedure, which decreased transiently during the initial abstinence period and finally returned to pre-abstinence levels after a period of 17–27 days. Acute withdrawal from CIE exacerbated irritability and anxiety in the animals. In addition, it was shown that in rats voluntarily taking alcohol (both after CIE and not subjected to this procedure), plasma corticosterone levels increased during drinking, with the same rats experiencing a decrease in the expression and signaling of glucocorticoid receptors in the medial prefrontal cortex during acute withdrawal. Interestingly, during prolonged abstinence, the expression of these receptors increases, and the rats show strong reinstatement of EtOH-seeking behavior. The authors of this study showed that the increased susceptibility to relapse in EtOH addiction may be partially due to altered expression of glucocorticosteroid receptors in the medial prefrontal cortex, suggesting that the transition to alcohol dependence may be accompanied by changes in stress-related pathways (exacerbating negative emotional symptoms) [51].

Therefore, AUD dependence impairs the negative feedback of the HPA axis, leading to increased levels of glucocorticoids in plasma, which in turn increases the activity of the TDO enzyme in the liver. In turn, TDO, like IDO, contributes to increased oxidation of TRP. The consequence of this is a reduction in the concentration of this amino acid in the liver, plasma, and brain, thus limiting the synthesis of 5-HT and redirecting the TRP metabolism towards the KP. This leads to the formation of neuroactive KP metabolites, which are additionally directly or indirectly involved in the biological mechanisms of AUD [38,52].

## 5. Kynurenine Pathway Enzymes

The association of KP metabolites with a number of diseases has led to significant efforts to therapeutically modulate the KP. It has been shown that modulation of the KP by inhibiting or stimulating the synthesis and/or activity of enzymes may constitute a new variant of the traditional therapy of neurological and psychiatric diseases. Particular attention is currently being paid to the development of inhibitors of the key enzymes of this pathway—IDO (IDO1, IDO2), TDO, and KMO [53]. Regulating KP activity based on these pharmacological actions also has great therapeutic potential for EtOH-dependent patients.

### 5.1. Indoleamine 2,3-Dioxygenase (IDO)/Tryptophan 2,3-Dioxygenase (TDO)

Breaking down TRP, IDO, and TDO limits the availability of this amino acid for the synthesis of 5-HT and influences the formation of neuroactive KP metabolites [54]. Numerous studies have shown the impact of increased expression of inflammatory mediators on the increase in the expression of IDO and TDO, cerebral 5-HT deficiencies, and the characteristic behavioral patterns in the form of cognitive (deteriorating memory function) and emotional (depression, anxiety) deficits [4,55]. The role of TDO in the brain was studied using, among other methods, TDO-knock-out (TDO-KO) mice. It has been reported that the central and peripheral concentrations of TRP and 5-HT and the concentration of TRP in the liver of TDO-KO mice were higher than in control ones [56,57,58,59]. The accumulation of TRP in the blood of TDO-KO mice induced an increase in serotonin synthesis, resulting in an observed improvement in cognitive and emotional functions. In these animals, a reduction in anxiety-like behavior and an increase in exploratory activity in behavioral tests (open-field and elevated plus maze tests) were observed [60,61,62]. Hattori et al. [63] conducted a comprehensive behavioral analysis (open-field test; light/dark transition test; elevated plus maze test; portlet forced swim test; startle response/prepulse inhibition test; social interaction test in a novel environment; Crawley’s sociability and preference for social novelty test; y-maze test; contextual and cued fear conditioning test) of TDO2-KO mice. The authors of this study showed slightly higher locomotor activity and lower anxiety behaviors in TDO2-KO mice. However, there was no effect of this mutation on other behaviors, such as prepulse inhibition, depression-like, social, and cognitive behaviors. The authors of the study explain the lack of clear phenotypes in this case by the lack of stress and inflammation that could induce TDO2 mRNA expression and suggest the need to conduct further research to clarify the role of this enzyme in behavioral phenotypes associated with mental disorders [63].

Defects in serotonin metabolism and abnormalities in the levels of serotonin and tryptophan in the blood of patients have been reported in many psychiatric disorders. It has also been shown that polymorphisms of the human TDO2 gene are associated with mental disorders such as depression, schizophrenia, ADHD, or substance abuse [64,65,66]. A clinical case of a patient with TDO deficiency accompanied by increased serotonin concentration in the blood (as a consequence of hypertryptophanemia) was also described. Medical examinations did not reveal any other obvious symptoms in this case, which allows us to assume that TDO inhibition, like IDO inhibition, may be a safe therapeutic target in the treatment of mood disorders. However, it is worth keeping in mind that hypertryptophanemia was diagnosed early in the patient and has been managed since childhood with a low-TRP diet, which may be important for the observations noted [67]. Therefore, further studies are needed to understand the metabolic consequences of TDO and IDO inactivation. This is particularly important in the context of the topic discussed because anxiety and depression are common symptoms associated with EtOH withdrawal and may cause disease relapses and treatment discontinuation. Patients with AUD who attempt to abstain from alcohol exhibit a variety of neuropsychiatric symptoms (depressed mood, anhedonia, anxiety, insomnia, loss of energy, and irritability) long after acute alcohol withdrawal has ended.

Characteristic relationships between the increase in TDO and IDO expressions, inflammation, and patterns of behavioral changes are also demonstrated by studies on animal models of AUD. Jiang et al. [16], using a mouse model of chronic drinking enriched with a grace period to replicate the alternating periods of withdrawal and relapse observed in the clinic, demonstrated the expected development of emotional deficits (depressive and anxiety behaviors) and memory deficits, which were explained by the observed nervous system inflammation and IDO activation. The authors of this study demonstrated overexpression of pro-inflammatory cytokines in the hippocampus, cerebral cortex, and amygdala of mice after alcohol consumption, as well as an increase in the expression of IDO1 mRNA and proteins in the mentioned brain structures. Unlike IDO1, IDO2 was expressed only at the mRNA level in the hippocampus and amygdala in this study. IDO2 is an IDO homolog with lower enzymatic activity than IDO1. However, TDO expression was not detected in any of the mentioned brain structures (neither at the mRNA nor protein levels). However, the authors do not exclude the possibility of strong expressions of IDO2 and TDO in other unexplored areas of the brain, nor the possible influence of these enzymes on the behavioral disorders observed in the model used. The use of an IDO inhibitor in the form of 1-methyl-L-tryptophan (1-MT) reversed the EtOH-induced changes in the form of increased KYN expression, decreased 5-HT expression, increased 3-HK/KYNA ratio in the brain (and the accompanying increase in QUIN concentration), and observed emotional deficits in mouse models [16]. This indicates the normalizing effect of IDO inhibition on changes induced by EtOH at many levels—biochemical, molecular, and behavioral. Other studies examining the early and late behavioral and biochemical effects of ethanol withdrawal in a rat model reported anxiety-like and depressive-like behaviors, respectively. Moreover, long-term EtOH withdrawal increased KYN levels, particularly in the prefrontal cortex, suggesting a possible link between increased IDO activity and depression-like reactions after long-term withdrawal [68]. Moreover, both increased IDO expression and increased levels of the cytokines IFN-γ, TNF-α, IL-1β, and IL-6 have been demonstrated in brain areas important in mediating depressive states [68,69]. Based on significant evidence for the activation of IDO and TDO enzymes in depression, inhibitors of these enzymes have been identified as therapeutic targets for the treatment of mood disorders [21,70].

### 5.2. Kynurenine 3-Monooxygenase (KMO)

The enzyme kynurenine 3-monooxygenase (KMO) catalyzes the hydroxylation reaction of KYN in the 3-position of the phenolic ring to produce 3-hydroxy-kynurenine (3HK), an endogenous generator of oxidative stress (promotes the production of reactive oxygen species in the respiratory chain), causing apoptosis of nerve cells. It is then transformed, under the influence of kynureninase, into hydroxyanthranilic acid (3-HANA), which has neurotoxic and immunosuppressive effects. 3-HANA is further converted into quinolinic acid (QUIN), an agonist of NMDA (*N*-methyl-d-aspartate) receptors. NMDA receptors play essential roles in numerous processes regulating cognitive functions (including neuroplasticity, memory formation, neuronal growth and survival, and learning processes related to the hippocampus), and numerous studies have shown a strong connection between these receptors and addiction. The neurotoxic effect of QUIN is linked, among other things, to stimulation of the NMDA receptor, which leads to inhibition of glutamate reuptake by astrocytes, thereby increasing its transmission at the synapse, called oxidative stress, and intensifying apoptosis of astrocytes [21].

Similarly to IDO, KMO is activated under the influence of pro-inflammatory cytokines. However, this is critical for further transformations of the KP towards synthesizing toxic metabolites. KYN is a branching point of the KP, and alternatively, it can be metabolized into KYNA, which has a neuroprotective effect. Researchers dealing with the neurobiology of neurological and psychiatric diseases point to the importance of the disturbed balance in KP metabolism, consisting of the predominance of the synthesis of neurotoxic metabolites 3HK and QUIN at the expense of the neuroprotective KYNA. KYNA is an NMDA receptor antagonist and thus can reduce glutamatergic transmission. The inhibition of KMO may be a therapeutic target in neurological and psychiatric diseases, including the treatment of AUD. Recent research has shown that EtOH-induced behavior can be performed by modifying KP metabolism and redirecting it in the way KYN is used. A study by Giménez-Gómez et al. [71] showed that KMO inhibition with Ro 61-8048 reduces voluntary EtOH intake and preference in mice subjected to drinking in the dark (DID) and two-bottle choice paradigms, respectively. The association of this effect with increased KYN concentration is indicated by the relationship between intraperitoneal administration of KYN and the reduction in excessive EtOH consumption, demonstrated in the DID model. Interestingly, Ro 61-8048 did not change plasma acetaldehyde concentrations but prevented EtOH-induced dopamine release in the nucleus accumbens shell, suggesting a critical involvement of the reward system in the reduction in EtOH intake induced by increases in KYN and KYNA concentrations [71]. Other studies have shown a similar effect. Gil de Biedma-Elduayen et al. [72] observed that KMO inhibition using Ro 61-8048 reduced EtOH intake and preference in mice of both sexes exposed to the CIE model and could be prolonged by blocking KYN efflux from the brain. The authors of this study showed that this is related to the influx of KYN from the periphery to the brain. With a decrease in EtOH consumption, a significant increase in the concentration of KYN and KYNA in animals’ plasma and limbic forebrain was demonstrated. In the CNS, only about 40% of KYN is formed locally. In comparison, 60% is captured from the periphery, which allows us to conclude that changes in the levels of KYN metabolites in the brain result from changes in their peripheral concentrations (particularly since Ro 61-8048 does not cross the blood–brain barrier (BBB)). The described results thus demonstrate that modulation of the KP (through KMO inhibition) may be an effective pharmacological tool for modifying ethyl alcohol consumption [72].

## 6. Kynurenine Pathway Metabolites

KYN, like TRP and 3-HK, is absorbed into the brain through the BBB. About 40% of KYN is thought to be produced in the brain by the local breakdown of TRP, while about 60% of KYN is transported into the brain from the periphery by the large neutral amino acid transporter (LNAA). TRP and other branched-chain and aromatic amino acids compete for LNAA [21]. It has been shown that administering a mixture of these amino acids without TRP functionally blocks the central passage of TRP [73]. Thus, the availability of TRP to the human brain depends on the ratio of TRP to LNAA in plasma. 

In the brain, KYN can be metabolized by different cells by producing various neuroactive compounds. The synthesis of QUIN (which, in addition, can also enter the brain from the periphery), like the synthesis of 3-HK and its downstream metabolites (including 3-HAA), occurs in microglia and other cells of monocyte origin. KYNA, on the other hand, is synthesized in astrocytes, neurons, and oligodendrocytes [74,75]. However, human astrocytes can metabolize QUIN produced by neighboring microglia [21].

KYNA and QUIN are essential for the course of EtOH addiction because they can alter excitatory neurotransmission and mediate the neuroimmune system.

### 6.1. Kynurenic Acid

KYNA is an endogenous metabolite of TRP formed from KYN by the KYN aminotransferases (KAT) I and KAT II in the CNS and peripheral tissues. It is also the final product of KP metabolism, with no further metabolic transformation. Regulation of KYNA synthesis in the CNS and periphery is a complex phenomenon influenced by many factors. A key factor regulating KYNA synthesis is the supply of KYN as a precursor of KYNA, as detailed when discussing the earlier stages of the KP. At this stage of the transformation pathway, KAT enzyme activity is crucial for KYNA formation. Numerous studies have shown that the reduction in KYNA concentration is affected by the presence of aminooxyacetic acid (AOAA), which blocks KAT enzymes both after intracerebral and peripheral administration [76,77,78]. Brain KYNA is thought to be synthesized mainly by KAT II, and a recent study published by Rentschler et al. [79] using an inhibitor of this enzyme (PF-04859989) showed a marked reduction in KYNA levels in the hypothalamus, basal forebrain, hippocampus, cortex, and frontal cortex. On this basis, the authors of this paper postulate that most KYNA synthesis in these regions is KAT II-dependent. The central production of KYNA from KYN is significant because, unlike KYN, KYNA does not penetrate or cross the BBB in tiny amounts. From a psychopharmacological point of view, on the other hand, the critical fact is that KYNA is an endogenous substance that modulates neurotransmission, primarily in glutamatergic transmission. As a glutamate antagonist, it can inhibit its ionotropic receptors: the *N*-methyl-d-aspartate receptor (NMDA), the kainate receptor (KAR), and the alpha-amino-3-hydroxy-5-methyl-4-isoxazolopropionic acid receptor (AMPA). KYNA also acts as a negative allosteric modulator of the α7-nACh receptor [14,80,81].

The glutamatergic system, whose neurons constitute the most substantial group in the CNS, is essential from the perspective of the mechanisms responsible for the development of addiction. This system connects several brain structures, including efferent pathways from the cortex to the thalamus, nucleus accumbens, and amygdala [82,83,84]. These structures are part of the reward system, which is activated during the satisfaction of urges and when engaging in activities perceived as pleasurable [85,86,87]. Overactivity of this system occurs in substance addiction. Glutaminergic projections from the prefrontal cortex to the basal nuclei and hippocampus are also noteworthy. The glutamatergic system, connecting these brain circuits (related to memory and perception), is involved in the processes of remembering, learning, and consolidation of memory traces primarily in the neuronal mechanism of so-called long-term synaptic reinforcement [88,89,90,91]. This phenomenon occurs due to repeated stimulation of neurons by strong impulses, causing an intense release of glutamate from the synapse of the presynaptic neuron into the inter-synaptic space. It then interacts with NMDA and AMPA receptors, which triggers a cascade of reactions that accelerate the transmission of nerve impulses [92,93]. This phenomenon occurs primarily in the hippocampus, where the created memory trace becomes clearer [94,95]. In addition to the hippocampus, the highest density of NMDA receptors has been shown in the cerebral cortex, striatum, and amygdala [96]. Moreover, they are found in the CNS and the adrenal glands, pituitary gland, and pineal gland. A high density of NMDA receptors is therefore seen in the cortical and subcortical structures of the brain that make up the so-called limbic system responsible for regulating drive and emotional behavior, as well as in the endocrine glands that make up the HPA axis [97].

Clinical and preclinical studies have confirmed that changes in glutamatergic neurotransmission are related to substance abuse behavior and may contribute to the development of addiction and the risk of relapse [98,99,100]. They persist long after the substance has been eliminated from the body and its effects have subsided. Therefore, the impact of endogenous substances that are neuromodulators of glutamatergic transmission, which include KYNA, is interesting from the point of view of pharmacotherapy of addiction [101,102,103,104]. All the more so, studies have unequivocally confirmed the role of KYNA in the pathophysiology of psychiatric disorders in recent years. A meta-analysis by Marx et al. [105] involving more than 100 studies found a reduced KYNA to KYN ratio for mood disorders (major depressive disorder, bipolar disorder). The authors of this paper searched electronic databases for studies evaluating the metabolites involved in the KP (tryptophan, kynurenine, kynurenic acid, quinolinic acid, 3-hydroxykinurenine, and their associated ratios). The published results confirm the preferential metabolism of KYN to the potentially neurotoxic QUIN rather than the neuroprotective KYNA during these disorders. A similar meta-analysis by Ogyu et al. [106] showed an association of KP metabolites with the pathophysiology of depression, demonstrating lower levels of KYNA in depressed patients and elevated levels of QUIN in patients not on antidepressants. Another clinical study on the role of the KP in addictive disorders (ADD) considering a cohort of 99 young adults (30 AUD patients, 34 behavioral addiction patients, and 35 healthy controls) found higher stress levels, lower resilience, and impaired executive function in AUD patients compared to the healthy control group. Significantly, in the context of the topic, the AUD group showed considerably elevated KYN levels and KYN/TRP ratios and reduced KYNA levels and KYNA/KYN ratios compared to healthy controls [97]. Previous studies on the role of the KP in addiction are promising. They increasingly postulate the role of KYNA in counteracting the addictive effects of drugs, including EtOH (including through a mechanism related to the regulation of glutamatergic transmission), making it a potentially attractive target for pharmacological treatment of addiction (Figure 2).

### 6.2. Quinolinic Acid

QUIN is produced non-enzymatically by spontaneous conversion from the precursor 2-amino-3-carboxymuconic-6-semialdehyde (ACMS) in the absence of the competing enzyme 2-amino-3-carboxymuconic-6-semialdehyde decarboxylase (ACMSD). ACMSD can convert ACMS into picolinic acid, inhibiting QUIN neurotoxic effects. However, when the enzyme is saturated, inactive, or absent, QUIN is spontaneously formed instead. Although the detailed regulation of this enzyme has not yet been established, the influence of inflammation on the increase in ACMS levels has been demonstrated. Under conditions of increased inflammation, ACMSD saturation and increased ACMS levels may lead to increased QUIN production [74,107,108]. QUIN is then broken down by the enzyme quinolinate phosphoribosyl transferase (QPRT). QPRT occurs both in the CNS and in peripheral organs. Therefore, QPRT activity regulates QUIN concentration locally for a specific tissue. In the brain, QUIN is produced primarily by microglia cells and infiltrating macrophages, i.e., cells visible during neuroinflammation [74,108,109]. Numerous clinical studies indicate that QUIN may act directly or be accessory to neuronal dysfunction or death and disrupt the integrity of the BBB [110].

Several mechanisms have been described for QUIN’s neurotoxicity, and the best described involves the activation of NMDA receptors at pathophysiological concentrations. Activation of the NMDA receptor by QUIN has been confirmed, among other things, by abolishing the effect of this acid through the use of NMDAR antagonists [111,112,113,114]. In addition, QUIN has been shown to bind specifically to NMDA receptors containing NR2A and NR2B subunits. NMDA receptors are expressed mainly in the forebrain, and neurons in the hippocampus, striatum, and neocortex are the most sensitive to QUIN toxicity [74]. Activation of NMDA receptors stimulates glutamate release by neurons, inhibiting its uptake by astrocytes and astroglia glutamate synthesis. The consequence is the production of reactive oxygen species and a reduction in endogenous antioxidants [109,115]. In turn, following the disruption of energy metabolism, QUIN can exacerbate its toxicity and that of other excitotoxins in an additive or synergistic manner.

Bano et al. [116] showed that QUIN of peripheral origin plays a vital role in the behavioral manifestation of alcohol withdrawal in a rat model. The alcohol withdrawal group demonstrated an increase in QUIN activity in the liver, accompanied by a rise in TDO activity in the brain. At the same time, no significant changes in TRP levels in the brain were observed, with a substantial decrease in central 5HT levels. The above results suggest an increase in TRP turnover and a redirection of TRP metabolism toward the KP with the production of excitotoxic QUIN, accompanied by the pattern of the behavioral changes characteristic of the disorders in question [116]. Mechtcheriakov et al. [117] examined TRP and KYN metabolism during acute alcohol withdrawal in patients with AUD. In this study, they measured serum levels of TRP, KYN, QUIN, KYNA, and immune activation marker neopterin (NEO) on the first, fifth, and tenth days of alcohol withdrawal in patients with AUD. This study showed a significant effect of NEO and KYN on QUIN levels, but KYNA only affected KYN levels. This finding underscores the possibility that even modest immune activation may be associated with increased metabolic pathway activity toward QUIN formation [117].

Currently, there are a limited number of studies elucidating the role of QUIN in addiction. However, elevated peripheral and central QUIN levels have been found in psychiatric patients and preclinical models of psychiatric disorders. Patients with depression and suicidal tendencies have been found to have high levels of QUIN in the cerebrospinal fluid [118]. A correlation was also found between high QUIN levels and IL-6 interleukin in the cerebrospinal fluid of patients with multiple suicide attempts in the two years preceding the study [107]. It was shown that QUIN levels increased over time, and KYNA levels decreased in the cerebrospinal fluid of these subjects. The cited studies directly or indirectly confirm that QUIN may be one of the factors responsible for addiction, including AUD, and show a link between QUIN and the psychiatric disorders that often accompany addiction. Studies from recent decades on the biological causes of psychiatric disorders have identified increased metabolism of TRP to QUIN via the KP with associated serotonin deficiency, increased levels of QUIN leading to excessive NMDA signaling and potential disruption of BBB integrity, and increased levels of inflammatory factors (mainly IL-6), among others. The consequence is microglia dysregulation and reduced inhibition of glutamatergic neurons, resulting in increased glutamate release and excitotoxicity [119].

## 7. Conclusions

In recent years, discoveries have been providing biological insights into the role of the KP in neurological and psychiatric disorders, creating diagnostic and therapeutic opportunities for these diseases. AUD, along with depressive disorders, is among the most common psychiatric disorders. Moreover, they most often co-occur, which is associated with greater severity and a worse prognosis for both disorders. AUD patients who try to abstain from alcohol exhibit a variety of neuropsychiatric symptoms long after acute alcohol withdrawal is over. These prolonged symptoms, such as depressive mood, anhedonia, anxiety, insomnia, loss of energy, and irritability, may be a consequence of EtOH-related chronic glutaminergic hyperactivity. The studies described in this review point to an immune-related mechanism that may increase glutaminergic activity in AUD patients by altering the balance of KP metabolites, thereby potentially increasing the risk of relapse. A better understanding of these mechanisms offers a real chance to elucidate the biological basis of AUD and implement new therapeutic interventions for it in the future. These studies should include patients with AUD during acute EtOH withdrawal, as well as patients with AUD who have been abstinent for a long time, which may be necessary for understanding the mechanisms responsible for relapse that are inherent in the characteristics of AUD.

Further research should focus on the long-term effects of chronic immune system activation in AUD on neurodegenerative and neurobehavioral symptoms in the context of changes in the activity of KP-modulating enzymes and a wide range of KYN metabolites. Indeed, the enzymes that modulate KP activity and the metabolites of this pathway are targets for developing new strategies for managing AUD-related therapeutic interventions. KP activity can be modulated by inhibiting this pathway’s enzymes, primarily IDO, TDO, KMO, and ACMSD. Inhibitors of these enzymes have been shown to normalize EtOH-induced changes at biochemical, molecular, and behavioral levels. The most promising pharmacological action in this regard is to alter the activity of the KP by redirecting its pathway towards synthesizing the neuroprotective metabolite KYNA while reducing the production of the neurotoxic QUIN. This is related to the fact that KYNA and QUIN are NMDA receptor antagonists and agonists, respectively. NMDA receptors, in turn, modulate glutamatergic transmission. Since KYNA and QUIN have opposite effects on NMDAR, the QUIN/KYNA ratio may be an appropriate measure to suggest the overall level of NMDAR stimulation. In the literature, an increased QUIN/KYNA ratio is often referred to as a neurotoxic ratio. However, the effects of metabolites on the receptor are primarily agonistic/antagonistic before any neurotoxic effects occur. Thus, further studies verifying the interrelationships of the numerous intermediate metabolites of the KP and the enzymes that modulate the activity of this metabolism pathway, as well as the complex metabolic, neurobiological, and immunological mechanisms, are needed to understand the complex pathomechanisms of AUD. Nonetheless, it should be borne in mind that the biological determinants of addiction are very diverse and not fully understood. So far, research into the biological mechanisms of addictive disorders has not brought the expected spectacular breakthroughs in addiction treatment. However, it has made it possible to conceptualize addiction as a highly complex disorder and has set new directions for research. Moreover, the practical problems of translating preclinical and clinical knowledge into new effective therapeutic interventions show that the road from research to actual medical applications is long and unpredictable, and even the most promising discoveries can require large amounts of resources and time to lead to practical applications.

## Figures and Tables

**Figure 1 ijms-25-04845-f001:**
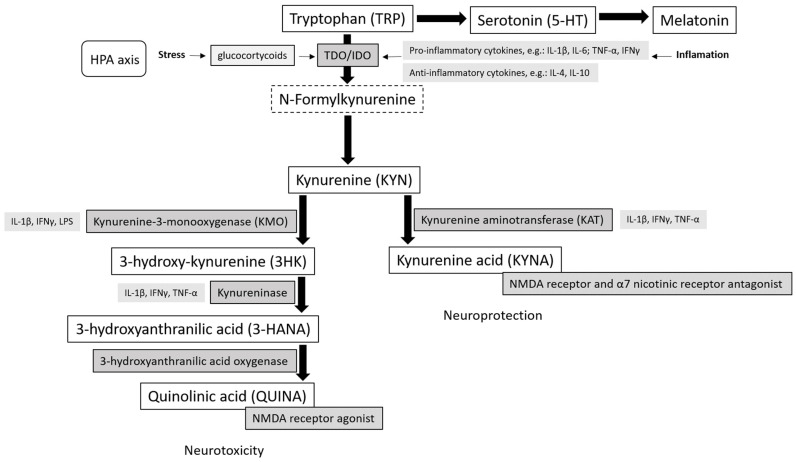
Tryptophan metabolism.

**Figure 2 ijms-25-04845-f002:**
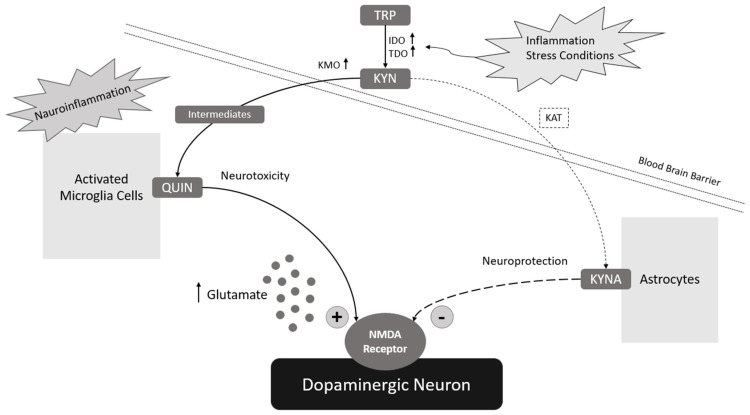
The role of KP metabolites in AUD (preferential metabolism of KYN to the potentially neurotoxic QUIN rather than the neuroprotective KYNA during addictive disorders).

## Data Availability

Not applicable.

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
