# Peer review of "Implications of Kynurenine Pathway Metabolism for the Immune System, Hypothalamic–Pituitary–Adrenal Axis, and Neurotransmission in Alcohol Use Disorder"

_ijms, 2024, doi:10.3390/ijms25094845_

Round 1

Reviewer 1 Report

Comments and Suggestions for Authors

I read with interest this review on the kynurenine pathway and alcohol use disorders. I thought this paper was generally well organized.

I consider it of sufficient quality to publish as it is now. However, if I may venture to raise a concern, I was concerned about the lack of adequate use of figure. In particular, everything after Figure 1 is explained in text. Readers with prior knowledge of the Kynurenine pathway will have some understanding, but for others it will be a little bit difficult.

This paper would be more improved if authors could summarize in a graphical abstract (figure 2), especially the areas of action of kynurenine-related metabolites (KYNA and QUIN) in the brain (especially reward system-related areas), their protective/excitotoxic effects, their relation to alcohol use disorders including HPA axis, etc., and their relation to glial cells. 

Author Response

Thank you for taking the time and effort to review the manuscript. We sincerely appreciate your valuable comments and suggestions, which helped us improve the quality of the manuscript. As per your suggestion, we have added a figure (LL422-426) showing the role of kynurenine pathway metabolites in alcohol use disorder (the preferential metabolism of KYN to the potentially neurotoxic QUIN (in activated microglia cells) rather than the neuroprotective KYNA) including through a mechanism related to the regulation of glutamatergic transmission and interaction with NMDA receptors (QUIN and KYNA as NMDA receptor agonist and antagonist, respectively).

Reviewer 2 Report

Comments and Suggestions for Authors

The presented work shows interest in the role of the kynurenine pathway (KP) in mechanisms related to addictive behaviors.
The authors present numerous reports from the literature that indicate the relationship of KP metabolism with the immune system, the hypothalamic-pituitary-adrenal (HPA) axis and neurotransmission in the central nervous system.
Literature analysis of the presented studies shows patterns of associations with alcohol use (AUD) and the neuroendocrine and immune systems, which may be important for a better understanding of the complex phenomenon of addiction to psychoactive substances and alcohol.
In the conclusions of the literature review, the authors considered the most promising to be the possibility of modulating the activity of KP by redirecting its pathway towards the synthesis of the neuroprotective metabolite KYNA while reducing the production of the neurotoxic QUIN.

The work is well written, although the conclusion fails to mention that the problem of addiction is a complex interaction of many neurotransmitters and the enzymes that metabolize them, so the presented influence of the kynurenine pathway (KP) should be approached with caution.    

Author Response

Thank you for taking the time and effort to review the manuscript. We sincerely appreciate your valuable comments and suggestions, which helped us improve its quality. Following your suggestion, we have supplemented the last paragraph in Conclusions (LL525-535) with information on conceptualizing addiction as a highly complex disorder.

Alcohol use disorder (AUD), understood as a severe chronic CNS disease, is characterized by the interplay of numerous mechanisms occurring at different levels of the body's functioning. Therefore, in this review, we have presented the importance of kynurenine pathway (KP) in AUD in the context of critical systems from the point of view of this disease, i.e., the immune system, the physiological stress response (activation of the HPA axis), the reward system and neurotransmission.

Of course, the best-described neurobiological determinant of the addiction phenomenon is changes within the dopaminergic system. Although many years of research on dopamine have not brought the expected spectacular breakthroughs in addiction treatment, they have allowed another conceptualization of addiction as a very complex disorder and set new directions for research. Clinical and preclinical studies in recent years, in turn, have confirmed that changes in glutamatergic neurotransmission are related to substance abuse behavior and may contribute to the development of addiction and the risk of relapse. Notably, these changes are permanent and persist long after the substance has been eliminated from the body and its effects have subsided. Therefore, the action of endogenous substances that are neuromodulators of glutamatergic transmission, including kynurenic acid (KYNA), has become attractive from the point of view of addiction pharmacotherapy.   

Research on the role of the kynurenine pathway in addiction is, of course, still limited, but the role of KYNA in counteracting the addictive effects of drugs (through, among other things, a mechanism related to the regulation of glutamatergic transmission) is increasingly being postulated, making it a potentially attractive target for pharmacological treatment of addiction. This review aims to highlight the importance of further studies verifying the interconnectedness of the numerous intermediate metabolites of KYNA and the enzymes that modulate the activity of this pathway, as well as the complex metabolic, neurobiological and immunological mechanisms that provide an opportunity to understand the complex pathomechanisms of addictive disorders.

Reviewer 3 Report

Comments and Suggestions for Authors

The topic of the present paper „Implications of kynurenine pathway metabolism with the immune system, hypothalamic-pituitary-adrenal axis and neuro- transmission in the alcohol use disorder”, the authors provided an up-to-date summary of information indicating the relationship between alcohol use disorder and kynurenine pathway, both in terms of changes in the activity of this pathway and modulation of this pathway as a possible pharmacological approach for the treatment of alcohol use disorder.

The manuscript is well written, and the text is easy to read, but further studies verifying the interrelationship of the numerous intermediate metabolites of kynurenine pathway and the enzymes that modulate the activity of this metabolism pathway and the complex metabolic, neurobiological and immunological mechanisms are needed to understand the complex patho-mechanisms of alcohol use disorder.

The subsections are well discussed:

-         Tryptophan metabolism (Kynurenine pathway);

-         Inflammation and implications of kynurenine pathway metabolism;

-  hypothalamic-pituitary-adrenal axis and implications of kynurenine pathway metabolism;

    kynurenine pathway enzymes (Indoleamine 2,3-dioxygenase/ Tryptophan 2,3-dioxygenase; Kynurenine 3-monooxygenase);

     kynurenine pathway metabolites (Kynurenic acid, Quinolinic acid).

Finally, I conclude that:

-         the topic of the present manuscript is relevant on the field;

-   the introduction provides sufficient background and includes relevant references;

-         the reference list is recently and large.

Author Response

Thank you for taking the time and effort to review the manuscript. We sincerely appreciate your valuable comments and suggestions. The biological determinants of addiction are very diverse and, despite several decades of exploration, still need to be sufficiently understood. Alcohol use disorder (AUD), understood as a severe chronic CNS disease, is characterized by the interplay of many complex mechanisms occurring at different levels of the body's functioning. Therefore, in this review, we have presented the importance of the kynurenine pathway (KP) in AUD in the context of the critical systems involved in this disease, i.e. the immune system, the physiological stress response (activation of the HPA axis), the reward system and neurotransmission. Research on the role of the KP in AUD remains limited. Still, KYNA's role in counteracting the addictive effects of substances (including through a mechanism related to the regulation of glutamatergic transmission) is increasingly being postulated, making it a potentially attractive target for pharmacological treatment of addiction. This review aims to highlight the importance of further studies verifying the interconnectedness of the numerous intermediate metabolites of the KP and the enzymes that modulate the activity of this pathway, as well as the complex metabolic, neurobiological and immunological mechanisms that provide an opportunity to understand the complex pathomechanisms of addictive disorders.